# Ionic-liquid-facilitated transdermal absorption of lidocaine hydrochloride

**Xiaoting Ma[1], Yangzhe Zhou[2]\*, Ke Jiang[1], Yuqi Wen[1]\***

**1** Department of Traditional Chinese Medicine, Xiangya Hospital, Central South University, Changsha, China, **2** Department of Plastic surgery, Xiangya Hospital, Central South University, Changsha, China

\* zo7858667@gmail.com (YZ); wenyuqi1221@163.com (YW)

## Abstract

The increasing worldwide need for analgesics, along with patients' aversion to injectable anesthetics and limited compliance, highlights the necessity for noninvasive local anesthesia. However, the skin barrier poses a significant obstacle to the efficient transdermal administration of anesthetics. In this study, we synthesized a new ionic liquid (IL), tetradecylphosphonium hexanoate, by mixing sodium hexanoate and CYPHOS 101 at equimolar ratios. We systematically evaluated the capacity of this IL to improve the transdermal transfer of lidocaine hydrochloride through in vitro skin permeation tests and cytotoxicity investigations in HaCaT cells. The IL significantly enhanced the percutaneous absorption of lidocaine in a concentration-dependent manner. At 80 mM, the permeation efficiency increased by approximately 1.5 times compared with the control group, resulting in a significantly reduced onset time for topical anesthesia. While IL showed minimal cytotoxicity at low concentrations, higher concentrations were linked to increased cytotoxicity. ILs can effectively enhance the permeation of transdermal lidocaine. The ease of synthesizing ILs, their adjustable properties, and good compatibility make them promising for creating advanced topical anesthetics and broadening transdermal drug delivery applications.

## 1. Introduction

Pain can result from various diseases and injuries, significantly reducing quality of life and imposing substantial economic and social burdens [1–6]. For instance, the global prevalence of low back pain was estimated at 619 million in 2020, projected to reach 843 million by 2050, consistently ranking it as the primary cause of disability burden [7]. Lidocaine hydrochloride is a commonly used amide-type local anesthetic in clinical practice and serves as a Class Ib antiarrhythmic agent for treating ventricular arrhythmias. Its primary mechanism involves blocking voltage-dependent sodium channels, thereby inhibiting the generation and conduction of action potentials in nerves or cardiac muscles [8]. In anesthesiology, lidocaine provides a rapid onset, good penetration, and effective surface anesthesia for various clinical scenarios,

**Data availability statement:** All relevant data are within the manuscript and its Supporting information files.

**Funding:** This study was financially supported by the Innovative Research Group Project of the National Natural Science Foundation of China in the form of a grant (8207104259) received by YZ. This study was also financially supported by the Science and Technology Program of Hunan Province in the form of a grant (2019TP1001) received by YZ. This study was also financially supported by the Innovation-Driven Project of Central South University in the form of a grant (2020CX002) received by YZ. The funders had no role in study design, data collection and analysis, decision to publish, or preparation of the manuscript.

**Competing interests:** The authors have declared that no competing interests exist.

including infiltration, nerve block, and surface/mucosal anesthesia. Its potency and duration fall within a moderate range among commonly used local anesthetics, with favorable tissue diffusion owing to its physicochemical properties. Upon absorption into the bloodstream or intravenous administration, it exerts distinct dual excitatory and inhibitory effects on the central nervous system [8–13].

The most common clinical administration route for lidocaine hydrochloride is subcutaneous injection. However, this method is often accompanied by significant stinging and local discomfort, thereby reducing patient compliance [14–17]. To overcome this limitation, researchers have developed various alternative delivery strategies, including ultrasound-mediated transdermal delivery, electroporation, gel formulations, microneedles, and transdermal delivery [18–31]. Compared with other routes, transdermal delivery is considered advantageous owing to its noninvasiveness and avoidance of the first-pass effect, significantly improving drug bioavailability and patient experience [12,26,32]. However, the skin stratum corneum—the body's primary natural barrier—remains the main obstacle limiting effective transdermal drug absorption [33].

In recent years, ionic liquids (ILs) have attracted significant attention due to their distinct physicochemical properties and remarkable capacity to enhance skin permeability [34–39]. Comprising specific ratios of organic cations to organic/inorganic anions, IL typically possess melting points below 100 °C and remain liquid at room temperature, displaying outstanding solubility, thermal stability, and adjustability [34–39]. These attributes hold great promise for drug delivery applications. Mandal et al. significantly enhanced the transdermal delivery and therapeutic effectiveness of NFKBIZ siRNA by combining it with IL for psoriasis treatment [40]. Banerjee et al. utilized IL to enhance the oral absorption efficiency of insulin, presenting new opportunities for diabetes management [41]. Therefore, ILs are not only innovative transdermal permeation enhancers but also poised to serve as a vital platform for delivering complex drugs and biologics in the future.

In this study, we aimed to utilize a novel IL, tetradecylphosphonium hexanoate, synthesized from CYPHOS 101 and sodium hexanoate, to enhance the transdermal absorption of lidocaine hydrochloride. We hypothesized that this IL would disrupt the stratum corneum barrier and increase drug permeation while maintaining acceptable biocompatibility. To test this hypothesis, we systematically evaluated preliminary safety via cytotoxicity studies in HaCaT keratinocytes and assessed the permeation-enhancing efficacy through in vitro skin permeation assays.

## 2. Materials and methods

### 2.1 Materials

Sodium hexanoate (99% purity) was acquired from Aladdin. Trihexyltetradecylphosphonium (tetraalkylphosphonium) chloride (CYPHOS 101, purity ≥ 97%), lidocaine hydrochloride (purity: 99%), and silver nitrate solution were procured from Macklin. Phosphate buffer for cell culture was purchased from Procell. Dimethyl sulfoxide (DMSO) and MTT for cell proliferation assays were sourced from SolarBio. HaCaT cells were obtained from Xiangya Hospital of Central South University. All reagents were of analytical grade and used without additional purification.

## 2.2 Instruments

Transdermal testing utilized a Franz diffusion cell (TK-12D, Shanghai Kaikai). Characterization was performed using nuclear magnetic resonance (NMR) on a Bruker 400 MHz (Switzerland). Ultraviolet–visible (UV–vis) absorption spectra were obtained with a UV-2450 spectrophotometer (SHIMADZU, Japan). The Elx800 microplate reader (BioTek, U.S) was utilized.

## 2.3 Animal preparation

Male SD rats (weight: 180–220 g, age: 7–8 weeks) were provided by the Xiangya School of Medicine, Central South University(ethics audit number: CSU-2022–0170). All animals were housed under standard conditions (room temperature, 12-h light/dark cycle) with free access to a standard pellet diet and water. After a one-week acclimatization period, the rats were deeply anesthetized by an intraperitoneal injection of pentobarbital sodium (50 mg/kg). Euthanasia was subsequently performed via an overdose of the same anesthetic, and death was confirmed by the absence of breathing and cardiac arrest. The abdominal hair was then shaved, and the full-thickness abdominal skin was surgically excised. All subcutaneous fat and connective tissue were carefully removed. The excised skin samples were stored at −20 °C in a freezer and used within two weeks to preserve the integrity of the skin barrier for consistent in vitro permeation studies. The experiment was conducted in compliance with the Code of Ethics of the World Medical Association and approved by the Experimental Animal Welfare and Ethics Committee of Xiangya School of Medicine, Central South University.

## 2.4 Synthesis of the IL

First, sodium hexanoate (1.46 g, 0.011 mol) powder was dissolved in 15 mL of methanol. CYPHOS 101 (5.48 g, 0.011 mol) was then added to the solution and stirred for 15–20 min until fully mixed. Methanol was removed by rotary evaporation, and the IL was washed with deionized water until the washed water was chloride-free. IL was subsequently dried in a vacuum oven at 80 °C for 48 h. For the preparation of the lidocaine hydrochloride-IL, lidocaine hydrochloride powder was added to the IL to achieve a concentration of 1 mg/mL and thoroughly mixed.

## 2.5 Detection of lidocaine hydrochloride

Lidocaine hydrochloride was prepared at concentrations of 2, 3, 4, 6, 8, and 10 µg/mL. Subsequently, 100 µL of each concentration was introduced into the cuvette for analysis with a UV spectrometer.

## 2.6 Cytotoxicity test

To evaluate the cytotoxicity of IL, HaCaT cells were selected as the model due to their suitability in mimicking the human epidermal state [42]. First, $1 \times 10^4$ cells in 100 µL of culture medium were seeded into each well of a 96-well plate and incubated overnight. Upon confirming cell adhesion, the culture medium was aspirated and replaced with varying concentrations of IL (30, 50, and 80 mM) for 5 h. Subsequently, the medium was exchanged with 100 µL of Dulbecco's modified Eagle's medium (DMEM) and incubated for an additional 19 h (24 h in total). The control group was treated solely with DMEM throughout the experiment. Following this, 20 µL of MTT solution (0.5 mg/mL) was added to each well and incubated for 4 h. After discarding the MTT solution, 150 µL of DMSO was introduced, and the plate was agitated for 20 min on a balanced shaker. The absorbance was measured at 490 nm using a microplate reader.This assay evaluated in vitro cellular biocompatibility and was not designed to determine in vivo lethal dose (LD50) parameters.

## 2.7 In vitro skin permeation assays

The IL at concentrations of 30, 50, and 80 mM was combined with lidocaine hydrochloride (at a fixed concentration of 1 mg/mL) and assessed for transdermal absorption efficiency. The transdermal testing apparatus, as illustrated in Fig 1,

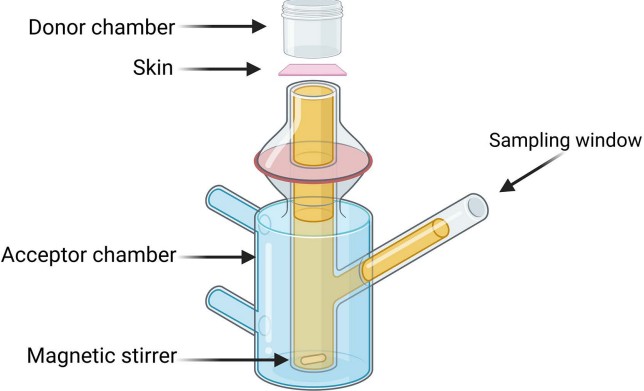

**Fig 1. Schematic of the Franz diffusion cell setup for in vitro skin permeation studies.** Schematic illustration of the Franz diffusion cell used for transdermal permeation experiments, showing donor, receptor, and skin membrane compartments. Created with BioRender.com.

consisted of a donor chamber positioned at the top, SD mouse skin treated in the middle (effective diffusion area: 3.14 cm²), and an acceptor chamber at the bottom with a capacity of 8 mL.

First, a small magnet was positioned in the acceptor chamber, with skin placed between the acceptor and donor chambers. The stratum corneum (SC) of the skin was elevated, and rat skin was slightly stretched to minimize wrinkles. Subsequently, 8 mL of phosphate-buffered saline (PBS) was added to the acceptor chamber. Particular attention was paid to the discharge of subcutaneous air bubbles; otherwise, the experimental results would have been affected. Then, 3 mL of IL/lidocaine hydrochloride solution with different IL concentrations was added to each donor chamber. The donor chamber was sealed with a film containing small holes to reduce evaporation. At regular intervals, 200 µL of the sample was removed from the sampling port and supplemented with 200 µL of PBS to keep the volume of the acceptor chamber constant. The chamber is placed in a magnetic box at a constant temperature of 37 °C for experiments.

### 2.8 Statistical analysis

Data are presented as mean ± standard deviation (SD) of at least three independent experiments. For skin permeation and cell viability studies, statistical significance was determined using one-way analysis of variance (ANOVA) followed by Tukey's post hoc test for multiple comparisons. A p-value of less than 0.05 was considered statistically significant. All analyses were performed using GraphPad Prism software.

## 3. Results

### 3.1 Characterization of the IL

The synthesized IL appeared as a clear, slightly yellow solution (Supporting Information, S1 Fig). The NMR spectrum of the IL was consistent with previous reports (Supporting Information, S2 Fig) [43]: 1H NMR (400 MHz, Chloroform-d) δ 2.53 (s, 1H), 2.43 (ddt, J = 13.0, 9.6, 5.4 Hz, 2H), 1.63 (dd, J = 17.3, 5.4 Hz, 0H), 1.47 (t, J = 4.9 Hz, 0H), 1.45–1.25 (m, 7H), 0.87 (d, J = 3.3 Hz, 0H); and 13C NMR (101 MHz, Chloroform-d) δ 179.22, 38.40, 32.09, 31.76, 30.94, 30.70, 30.55, 30.37, 30.22, 29.48, 29.38, 29.19, 28.83, 26.47, 22.55, 22.52, 22.27, 22.20, 21.68, 19.06, 18.59, 13.99, 13.96, and 13.77.

### 3.2 Detection of lidocaine hydrochloride

As shown in Fig 2, the absorbance intensity of the lidocaine hydrochloride solution at approximately 196 nm increased with increasing lidocaine hydrochloride concentrations within the 2–10 µg/mL range, exhibiting a correlation coefficient of

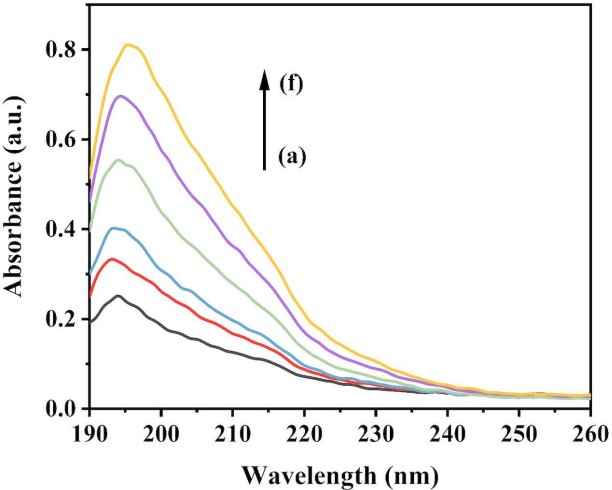 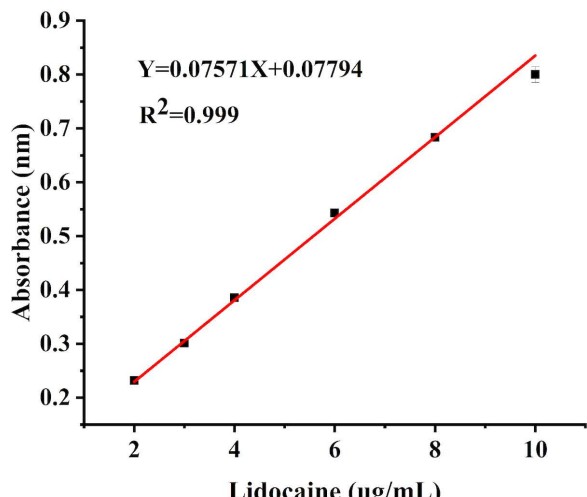

**Fig 2. UV–Vis calibration of lidocaine hydrochloride.** UV absorption spectra of lidocaine hydrochloride at different concentrations (a–f: 2, 3, 4, 6, 8, and 10 µg mL⁻¹) and calibration curve showing the linear relationship between absorbance at 196 nm and lidocaine concentration.

0.999. These absorbance readings were subsequently employed in a subsequent experiment to evaluate the transdermal administration of lidocaine hydrochloride.

### 3.3  HaCaT cell viability evaluation of the IL

The cytotoxicity of the IL towards HaCaT keratinocytes was evaluated, and the results are presented in Fig 3. The cell viability relative to the untreated control was $89.3 \pm 1.9\%$ for the 30 mM IL group and $88.9 \pm 3.1\%$ for the 50 mM IL group. There was no statistically significant difference between these two groups ($p > 0.05$, Tukey's test). In contrast, exposure to 80 mM IL significantly reduced cell viability to $41.2 \pm 0.8\%$ of the control level ($p < 0.001$ versus the control, 30 mM, and 50 mM groups; one-way ANOVA with Tukey's post hoc test).

### 3.4  In vitro skin permeation assays

As shown in Fig 4, the cumulative permeation of lidocaine hydrochloride increased over time and with increasing IL concentration, approaching a plateau at approximately 4 h. Compared with the control group, IL-treated groups exhibited significantly greater permeation at later time points. At 300 min, cumulative permeation in both the 50 mM and 80 mM IL groups was significantly higher than that in the control group ($p < 0.001$). No significant difference between the 50 mM and 80 mM groups was observed up to 3 h ($\leq 180$ min; ns). In contrast, the 80 mM group showed significantly higher permeation than the 50 mM group at 240 min ($p < 0.001$) and 300 min ($p < 0.01$). At 5 h, permeation in the 80 mM group was approximately 1.5-fold higher than that in the control group.

## 4.  Discussion

This study demonstrates that the synthesized IL significantly enhanced the transdermal permeation of lidocaine hydrochloride in a concentration-dependent manner. The 1.5-fold increase in cumulative permeation achieved with 80 mM IL suggests a strong potential to shorten the onset time of topical anesthesia, addressing a key clinical need for rapid-acting, non-invasive analgesics. We identified 50 mM as a practical concentration for further development because it improved permeation while maintaining high viability in HaCaT cells, suggesting a favorable balance between efficacy and biocompatibility. Although rapid onset is desirable for topical anesthesia, our permeation data were obtained

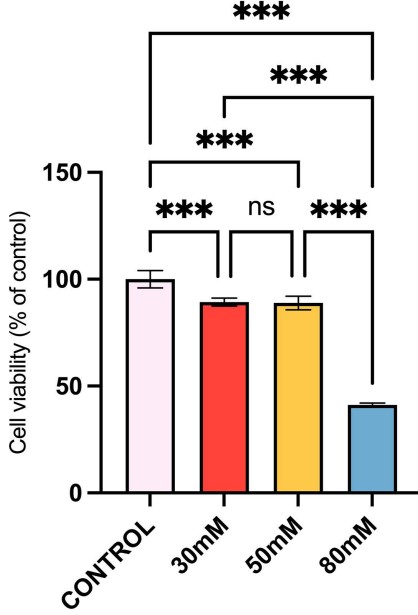

**Fig 3. Cytotoxicity of ionic liquids in HaCaT cells.** Viability of HaCaT cells treated with different concentrations of ionic liquids (ILs). After 5 h of IL exposure, the medium was replaced with fresh DMEM and cells were incubated for an additional 19 h (24 h total). Cell viability was assessed using the MTT assay. Data are presented as mean ± SD (n = 5). ***$p < 0.001$ versus the control group (one-way ANOVA with Tukey's post hoc test).

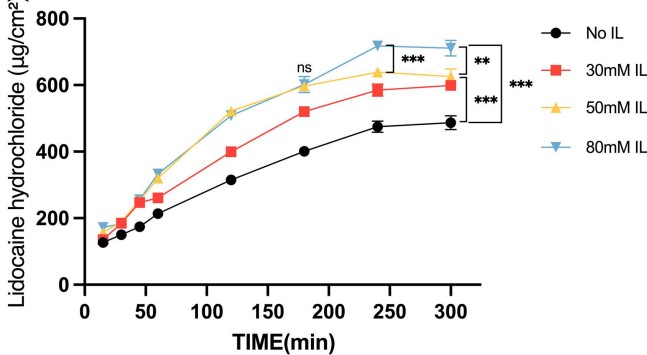

**Fig 4. Effect of ionic liquid concentration on lidocaine permeation.** Cumulative permeation of lidocaine hydrochloride over time in the presence of different concentrations of ionic liquids (ILs). Data are presented as mean ± SD (n = 3). Statistical comparisons were performed at each time point using one-way ANOVA followed by Tukey's multiple-comparisons test. Asterisks indicate significant differences between groups at the corresponding time point (**$p < 0.01$, ***$p < 0.001$); ns, not significant. For clarity, only key time points are annotated in the figure.

using an ex vivo mouse-skin model and therefore cannot be directly translated into clinical waiting time. Future studies should validate anesthetic efficacy and systemic safety in vivo, assess irritation and barrier effects in human-relevant skin platforms (e.g., reconstructed human epidermis), and evaluate long-term formulation stability. Overall, these results provide a foundation for developing IL-facilitated transdermal delivery systems for local anesthetics and potentially other therapeutics.

## 5. Conclusions

In this study, a novel tetraalkylphosphonium hexanoate IL was successfully synthesized. Cytotoxicity testing showed that the IL exhibited minimal toxicity toward HaCaT keratinocytes at 50 mM, supporting its biocompatibility at an effective working concentration. Building on this safety profile, in vitro skin permeation studies demonstrated that the IL significantly enhanced the transdermal permeation of lidocaine hydrochloride in a concentration-dependent manner, with an approximately 1.5-fold increase at 80 mM compared with the control. Collectively, these results suggest that this IL is a promising and biocompatible permeation enhancer for topical lidocaine delivery. Future work should focus on developing a practical transdermal formulation and conducting preclinical in vivo studies to evaluate lidocaine exposure, efficacy, and systemic safety.

## Supporting information

**S1 Fig. Appearance of the synthesized ionic liquid (IL) Photograph showing the physical state of the synthesized ionic liquid.** The IL presents as a clear and slightly yellow solution.
(TIF)

**S2 Fig. Nuclear Magnetic Resonance (NMR) spectrum of the synthesized ionic liquid (IL) NMR spectrum of the target synthesized ionic liquid.** The characteristic resonance peaks observed in the spectrum are consistent with those reported in previous studies, which confirms the successful synthesis and structural integrity of the IL.
(TIF)

**S3 Data. Raw data underlying all figures and statistical analyses reported in this study.**
(XLSX)

## Acknowledgments

The authors thank colleagues for their technical assistance and valuable discussions.

## Author contributions

**Conceptualization:** Yangzhe Zhou.

**Data curation:** Xiaoting Ma, Yuqi Wen.

**Formal analysis:** Xiaoting Ma, Yuqi Wen.

**Funding acquisition:** Yangzhe Zhou.

**Investigation:** Xiaoting Ma, Yangzhe Zhou.

**Methodology:** Ke Jiang.

**Project administration:** Yuqi Wen.

**Resources:** Yangzhe Zhou.

**Supervision:** Yuqi Wen.

**Validation:** Xiaoting Ma.

**Visualization:** Xiaoting Ma.

**Writing – original draft:** Xiaoting Ma, Yangzhe Zhou, Ke Jiang, Yuqi Wen.

**Writing – review & editing:** Xiaoting Ma, Yangzhe Zhou, Ke Jiang, Yuqi Wen.

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
