## [Decision Letter · Decision Letter 0]

21 Dec 2025

Dear Dr. Wen,

Thank you for submitting your manuscript to PLOS ONE. After careful consideration, we feel that it has merit but does not fully meet PLOS ONE’s publication criteria as it currently stands. Therefore, we invite you to submit a revised version of the manuscript that addresses the points raised during the review process.

We look forward to receiving your revised manuscript.

Kind regards,

Anisha DSouza, Ph.D (Tech)

Academic Editor

PLOS One

Journal Requirements:

3. To comply with PLOS One submissions requirements, in your Methods section, please provide additional information regarding the experiments involving animals and ensure you have included details on (1) methods of sacrifice, (2) methods of anesthesia and/or analgesia, and (3) efforts to alleviate suffering.

4. We note that this submission includes NMR spectroscopy data. We would recommend that you include the following information in your methods section or as Supporting Information files:

1) The make/source of the NMR instrument used in your study, as well as the magnetic field strength. For each individual experiment, please also list: the nucleus being measured; the sample concentration; the solvent in which the sample is dissolved and if solvent signal suppression was used; the reference standard and the temperature.

2) A list of the chemical shifts for all compounds characterised by NMR spectroscopy, specifying, where relevant: the chemical shift (δ), the multiplicity and the coupling constants (in Hz), for the appropriate nuclei used for assignment.

3)The full integrated NMR spectrum, clearly labelled with the compound name and chemical structure.

We also strongly encourage authors to provide primary NMR data files, in particular for new compounds which have not been characterised in the existing literature. Authors should provide the acquisition data, FID files and processing parameters for each experiment, clearly labelled with the compound name and identifier, as well as a structure file for each provided dataset. See our list of recommended repositories here: https://journals.plos.org/plosone/s/recommended-repositories

“This work was supported by the National Science Foundation of China (8207104259), the Hunan Provincial Science and Technology Plan Project, China (No. 2019TP1001), and the Innovation Driven Project of Central South University (2020CX002).”

6. Please note that funding information should not appear in any section or other areas of your manuscript. We will only publish funding information present in the Funding Statement section of the online submission form. Please remove any funding-related text from the manuscript.

7. Thank you for stating the following in the Acknowledgments Section of your manuscript:

“This work was supported by the National Science Foundation of China (8207104259), the Hunan Provincial Science and Technology Plan Project (No. 2019TP1001), and the Innovation Driven Project of Central South University (2020CX002); we also thank all team members for their assistance with the 11 experiment and data analysis.”

“This work was supported by the National Science Foundation of China (8207104259), the Hunan Provincial Science and Technology Plan Project, China (No. 2019TP1001), and the Innovation Driven Project of Central South University (2020CX002).”

8. Please include captions for your Supporting Information files at the end of your manuscript, and update any in-text citations to match accordingly. Please see our Supporting Information guidelines for more information: http://journals.plos.org/plosone/s/supporting-information ..

9. We notice that your supplementary figures are uploaded with the file type 'Figure'. Please amend the file type to 'Supporting Information'. Please ensure that each Supporting Information file has a legend listed in the manuscript after the references list.

Additional Editor Comments:

Dear Authors,

Thank you for submitting your manuscript to our journal. We have now received the reviewers’ reports, which are attached below for your reference. Based on these comments, we invite you to revise your manuscript accordingly. The reviewers have raised several points that require careful consideration, and we believe that addressing them will significantly strengthen the quality and clarity of your work.

Looking forward to receiving your revised manuscript.

Thanks!

Reviewer's Responses to Questions

**Comments to the Author**

1. Is the manuscript technically sound, and do the data support the conclusions?

Reviewer #1: Partly

Reviewer #2: Yes

2. Has the statistical analysis been performed appropriately and rigorously?

Reviewer #1: Yes

Reviewer #2: No

3. Have the authors made all data underlying the findings in their manuscript fully available?

Reviewer #1: Yes

Reviewer #2: Yes

4. Is the manuscript presented in an intelligible fashion and written in standard English?

Reviewer #1: No

Reviewer #2: Yes

Reviewer #1: regarding the manuscript titled "Ionic-liquid-facilitated transdermal absorption of lidocaine hydrochloride"

The introduction is well-written, justifying the addition of IL to local anaesthesia to improve the onset of action. at the end of the introduction, the authors wrote, "Skin permeation tests conducted in vitro confirmed the enhanced transdermal absorption of lidocaine hydrochloride facilitated by the IL. At an IL concentration of 80 mM, the transdermal absorption efficiency of lidocaine hydrochloride increased by 1.5 times, significantly reducing the wait time for epidermal anesthesia"

this is a conclusion, not an introduction.

material and method

- The material used was identified. The device used in the experiment – is it possible to add a picture for it?

- SD rats should be declared male or female to exclude any effects of sex hormones on transdermal application. and if the authors used both male and female, they should have mentioned if there were statistical difference or not.

- The authors mentioned that "The skin samples were then stored at -20℃ in a freezer and utilised within a two-week timeframe." Why did they store it? If the skin was used invitro skin permeation assays, they should mention that.

- This experiment has in vivo results?

- The authors did a toxicity test, and this is a great step, but they did not inform us about ED50 and LD50.

discussion: It is better to seperate results from discussion to compare your results with others if present in the discussions.

Reviewer #2: The authors have done a good job with designing experiments and drafting a technically sound manuscript. I would suggesthere are some statistics added to the data generated (if possible). I have attached a pdf with annoted comments which need to be addressed before the article gets published. Good job!

**Do you want your identity to be public for this peer review?** For information about this choice, including consent withdrawal, please see our For information about this choice, including consent withdrawal, please see our Privacy Policy .

Reviewer #1: No

Reviewer #2: **Yes:** Shashank BhangdeShashank Bhangde

---

## [Author Response · Author response to Decision Letter 1]

13 Jan 2026

Response to Reviewers

Manuscript ID: PONE-D-25-63253

Title: Ionic-liquid-facilitated transdermal absorption of lidocaine hydrochloride

Dear Academic Editor and Reviewers,

Thank you for the opportunity to revise our manuscript and for the constructive comments provided by the reviewers. We have carefully addressed all points raised and have made substantial revisions to improve the manuscript. Our point-by-point responses are detailed below. All changes in the manuscript have been highlighted using the “Track Changes” function.

Responses to Reviewer #1:

We sincerely thank Reviewer #1 for the thorough review and valuable suggestions.

1. Comment: The introduction contains a conclusion statement. “Skin permeation tests conducted in vitro confirmed…” should not appear in the introduction.

Response: We agree and have revised the final paragraph of the Introduction section. The conclusive statements have been removed and replaced with a clear statement of the study’s aims and hypothesis.

2. Comment: Please consider adding a picture of the Franz diffusion cell setup.

Response: A schematic diagram of the Franz diffusion cell setup is already provided as Fig. 1 in the original submission. We have now extended the title of this chart.

3. Comment: The sex of SD rats should be specified to exclude potential hormonal influences.

Response: We have specified that male SD rats were used in the revised Section 2.3 (Animal preparation).

4. Comment: Please clarify why skin samples were stored at −20 °C and used within two weeks.

Response: An explanation has been added to Section 2.3: skin samples were stored at −20 °C to preserve the integrity of the skin barrier and were used within two weeks to ensure experimental consistency in the in vitro permeation studies.

5. Comment: Does this study include in vivo results?

Response: This study is limited to in vitro evaluations. We have acknowledged this as a key limitation in the new Section 4. Discussion (Limitations and future perspectives).

6. Comment: Please provide ED50/LD50 data from the toxicity test.

Response: The cytotoxicity assay was conducted on HaCaT cells to assess in vitro biocompatibility and was not designed to determine in vivo lethal doses (LD50). We have clarified this point in the Section 2.7 (Cytotoxicity test) and in the Discussion.

7. Comment: It is better to separate the Results and Discussion sections.

Response: As suggested, we have restructured the manuscript by separating the combined “Results and discussion” section into two distinct chapters: “3. Results” and “4. Discussion.”

Responses to Reviewer #2:

We sincerely thank Reviewer #2 (Dr. Shashank Bhangde) for their thorough review, positive feedback, and constructive suggestions, which have significantly strengthened our manuscript.

1. Comment: Consider adding statistical analysis to the data where possible.

Response: We have performed comprehensive statistical analysis as suggested. A new Section 2.8 (Statistical analysis) has been added, describing the use of one-way ANOVA followed by Tukey’s post hoc test. The error bars in Fig. 3 and Fig. 4 now represent the standard deviation (SD), and statistically significant differences are indicated with asterisks (**p < 0.01, ***p < 0.001). The results sections (3.3 and 3.4) now include explicit statements of statistical significance.

2. Comment: The specific term “analgesic drug delivery” was missing.

Response: We thank the reviewer for highlighting this important omission. To accurately frame the therapeutic focus of our work and enhance the manuscript’s discoverability, we have added “analgesic drug delivery” as a keyword in the revised manuscript.

3. Comment: All citations appear at the end of the statement.

Response: We have reviewed and revised the manuscript to ensure all citations are placed at the end of their respective statements for consistency, including in the Introduction where general statements about lidocaine and ionic liquids were lacking support.

4. Comment: In current clinical setting, anesthesia is giving SC. You have only shown 1.5x increase in comparison to lidocaine given transdermally (which is not clinical standard). Rephrase this line.

Response: We thank the reviewer for this crucial clarification. We have rephrased the relevant line in the Conclusions section。

5. Comment: General suggestion to add future research directions.

Response: We appreciate this suggestion. As guided by the Editor, we have expanded the Conclusions to include a specific sentence on future work: “Future studies should focus on the formulation development of a practical transdermal delivery system and the preclinical evaluation of lidocaine bioavailability in vivo.”

6. Comment: “I have a little flow issue. The flow should be: synthesis – detection – cytotoxicity – skin permeation.”

Response: We agree with the reviewer’s suggestion for a more logical narrative flow. In the revised manuscript, we have reorganized the “3. Results” section accordingly:

3.1 Characterization of the IL

3.2 Detection of lidocaine hydrochloride

3.3 HaCaT cell viability evaluation of the IL

3.4 In vitro skin permeation assays

We believe this new order—presenting the material, then its safety profile, followed by its functional efficacy—provides a clearer and more compelling argument. All in-text citations to figures and results have been updated to reflect this new structure.We have also reordered the corresponding sections in ‘2. Materials and methods’ to maintain consistency with this improved narrative flow.

7. Comment:The issue of inconsistent units (ml, mg, µg)

Response: We thank the reviewer for identifying the unit errors. All instances of incorrect or ambiguous units have been identified and standardized throughout the manuscript to ensure accuracy and consistency.

Additional Revisions in Response to Journal Requirements:

We have also addressed all points raised by the Academic Editor to ensure full compliance with PLOS ONE policies.

1. Style Requirements: The manuscript has been adjusted as per the requirements and the structure has been ensured to comply with the guidelines.

2. ORCID iD: The corresponding author’s ORCID iD is registered and validated in the submission system.

3. Animal Ethics Details: As requested, Section 2.3 has been expanded to include specific details on (1) the method of euthanasia (anesthetic overdose), (2) the method of anesthesia (intraperitoneal injection of pentobarbital sodium, 50 mg/kg), and (3) efforts to minimize suffering (all procedures performed under deep anesthesia).

4. NMR Data: Details regarding the NMR instrument (Bruker 400 MHz), magnetic field strength have been added to Section 2.2. The NMR spectrum is provided as Fig. S2 in the Supporting Information

5. & 6. & 7. Funding Information: As instructed, all funding-related text has been removed from the manuscript’s Acknowledgements section. The funding statement provided in the submission system remains accurate. The role of the funders is stated here, as requested for the cover letter: “The funders were involved in providing financial support for the experiment and in the decision to publish.”

8. & 9. Supporting Information: Captions for the Supporting Information files (Fig. S1 and Fig. S2) have been included at the end of the main manuscript file. The file types have been designated correctly in the submission system.

We believe the revised manuscript has been significantly improved and now fully meets the publication standards of PLOS ONE. We are grateful for the insightful comments and hope you find our revisions satisfactory.

Sincerely,

Yangzhe Zhou, Yuqi Wen

---

## [Decision Letter · Decision Letter 1]

29 Jan 2026

Dear Dr. Wen,

Thank you for submitting your manuscript to PLOS ONE. After careful consideration, we feel that it has merit but does not fully meet PLOS ONE’s publication criteria as it currently stands. Therefore, we invite you to submit a revised version of the manuscript that addresses the points raised during the review process.

We look forward to receiving your revised manuscript.

Kind regards,

Anisha DSouza, Ph.D (Tech)

Academic Editor

PLOS One

**Journal Requirements:**

**Additional Editor Comments:**

Thank you for your patience. We have received the reviewers’ comments. One reviewer has provided a few minor comments on the manuscript. We kindly ask that you address these and resubmit the revised manuscript.

Reviewers' comments:

Reviewer's Responses to Questions

**Comments to the Author**

Reviewer #1: All comments have been addressed

Reviewer #2: (No Response)

2. Is the manuscript technically sound, and do the data support the conclusions?

Reviewer #1: Yes

Reviewer #2: Yes

3. Has the statistical analysis been performed appropriately and rigorously?

Reviewer #1: Yes

Reviewer #2: Yes

4. Have the authors made all data underlying the findings in their manuscript fully available?

Reviewer #1: Yes

Reviewer #2: Yes

5. Is the manuscript presented in an intelligible fashion and written in standard English?

Reviewer #1: Yes

Reviewer #2: Yes

Reviewer #1: (No Response)

Reviewer #2: The authors have done a good job drafting the manuscript and addressing the comments in the first round of review. For that, I congratulate you.

There are minor comments that need to be addressed to improve the quality of your manuscript and to be accepted in this journal.

**Do you want your identity to be public for this peer review?** For information about this choice, including consent withdrawal, please see our For information about this choice, including consent withdrawal, please see our Privacy Policy .

Reviewer #1: No

Reviewer #2: **Yes:** Shashank BhangdeShashank Bhangde

---

## [Author Response · Author response to Decision Letter 2]

23 Feb 2026

Response to Reviewers

Manuscript ID: PONE-D-25-63253

Title: Ionic-liquid-facilitated transdermal absorption of lidocaine hydrochloride

Dear Academic Editor and Reviewers,

Thank you for the opportunity to revise our manuscript and for the constructive comments provided by the reviewers. We have carefully addressed all points raised and have made substantial revisions to improve the manuscript. Our point-by-point responses are detailed below. All changes in the manuscript have been highlighted using the “Track Changes” function.

Response to Reviewer #1

We appreciate Reviewer #1’s positive evaluation of our revised manuscript. We are pleased that all previous comments have been adequately addressed and that the manuscript is now considered technically sound, statistically rigorous, and clearly presented. Thank you for your careful review and support.

Response to Reviewer #2

We sincerely thank the reviewer for the positive evaluation and constructive suggestions. We appreciate your recognition of our efforts in addressing the previous round of comments. Below we provide a detailed point-by-point response to the minor comments raised in this round of review.

Comment 1: You cannot claim that since the in vitro model is a mouse skin and not human model or in vivo study. Please rephrase this.

Response

We agree with the reviewer that conclusions regarding clinical waiting time should not be directly inferred from an ex vivo mouse-skin model. We have revised the corresponding statement in the Discussion to avoid clinical overinterpretation. The text now clarifies that the enhanced permeation observed within the early time window was obtained under experimental conditions using ex vivo mouse skin and does not directly predict clinical onset time. We have also added a statement emphasizing the need for validation in human skin models and in vivo studies.

The revised text reads as follows:

“Although rapid onset is desirable for topical anesthesia, our permeation data were obtained using an ex vivo mouse-skin model and therefore cannot be directly translated into clinical waiting time.”⸻

Comment 2: Statistics were only shown for the 5-hour time point. Please clarify whether statistical analyses were conducted at each time point.

Response

We thank the reviewer for this important comment. Statistical comparisons were performed at each individual time point using one-way ANOVA followed by Tukey’s multiple-comparisons test. The revised Figure 4 now reflects the statistical outcomes more comprehensively.

Specifically:

• No significant difference was observed between the 50 mM and 80 mM groups up to 3 h (≤180 min), which is now explicitly indicated as “ns” in the figure.

• Significant differences between the 50 mM and 80 mM groups emerged at 240 min (***p < 0.001) and 300 min (**p < 0.01).

• At 300 min, both the 50 mM and 80 mM groups were significantly higher than the control group (***p < 0.001).

For clarity, only key time points are annotated in the figure, while statistical testing was conducted at all sampling times. The Results section (Section 3.4) and figure legend have been revised accordingly.

Comment 3: Improve clarity and organization of Introduction,Results and Discussion.

Response

We have revised Sections 1.(Introduction), 3.4 (Results) and 4 (Discussion) to improve logical flow and readability. In the Conclusions section, we reordered the presentation to follow the logical progression of cytotoxicity evaluation before permeation analysis, as suggested.

Comment 4: Minor spelling and typographical errors were noted.

Response

We thank the reviewer for carefully identifying minor spelling and typographical errors. All reported issues have been corrected in the revised manuscript. In addition, the entire manuscript has been thoroughly proofread to ensure consistency in terminology, grammar, and formatting.

We believe the revised manuscript has been significantly improved and now fully meets the publication standards of PLOS ONE. We are grateful for the insightful comments and hope you find our revisions satisfactory.

Sincerely,

Yangzhe Zhou, Yuqi Wen

---

## [Editor Report · Decision Letter 2]

26 Feb 2026

Ionic-liquid-facilitated transdermal absorption of lidocaine hydrochloride

PONE-D-25-63253R2

Dear Dr. Wen,

We’re pleased to inform you that your manuscript has been judged scientifically suitable for publication and will be formally accepted for publication once it meets all outstanding technical requirements.

Kind regards,

Anisha DSouza, Ph.D (Tech)

Academic Editor

PLOS One

Additional Editor Comments (optional):

Thank you for addressing the questions raised by the reviewers. The paper can be accepted in the current form.
---

## [Editor Report · Acceptance letter]

PONE-D-25-63253R2

PLOS One

Dear Dr. Wen,

I'm pleased to inform you that your manuscript has been deemed suitable for publication in PLOS One. Congratulations! Your manuscript is now being handed over to our production team.

Kind regards,

on behalf of

Dr. Anisha DSouza

Academic Editor

PLOS One